# The Association between the Severity of Distal Sensorimotor Polyneuropathy and Increased Carotid Atherosclerosis in Individuals with Type 2 Diabetes

**DOI:** 10.3390/diagnostics14171922

**Published:** 2024-08-31

**Authors:** Dong-Yi Hsieh, Yun-Ru Lai, Chih-Cheng Huang, Chi-Ping Ting, Wen-Chan Chiu, Yung-Nien Chen, Chia-Yi Lien, Ben-Chung Cheng, Ting-Yin Lin, Hui Ching Chiang, Cheng-Hsien Lu

**Affiliations:** 1Departments of Neurology, Kaohsiung Chang Gung Memorial Hospital, Chang Gung University College of Medicine, Kaohsiung 83301, Taiwan; b9202095@cgmh.org.tw (D.-Y.H.); yunrulai@cgmh.org.tw (Y.-R.L.); u9301024@cgmh.org.tw (C.-Y.L.); a0917604453@gmail.com (H.C.C.); 2Departments of Hyperbaric Oxygen Therapy Center, Kaohsiung Chang Gung Memorial Hospital, Chang Gung University College of Medicine, Kaohsiung 83301, Taiwan; 3Department of Neurology, Chi-Mei Medical Center, Tainan 73657, Taiwan; hjc2828@gmail.com; 4Departments of Nursing, Kaohsiung Chang Gung Memorial Hospital, Chang Gung University College of Medicine, Kaohsiung 83301, Taiwan; cipin@cgmh.org.tw (C.-P.T.); s971078@cgmh.org.tw (T.-Y.L.); 5Departments of Nursing, Meiho University, Pingtung 91202, Taiwan; 6Departments of Internal Medicine, Kaohsiung Chang Gung Memorial Hospital, Chang Gung University College of Medicine, Kaohsiung 83301, Taiwan; testercwt@gmail.com (W.-C.C.); casearchen@gmail.com (Y.-N.C.); benzmcl@gmail.com (B.-C.C.); 7Center for Shockwave Medicine and Tissue Engineering, Kaohsiung Chang Gung Memorial Hospital, Chang Gung University College of Medicine, Kaohsiung 83301, Taiwan; 8Department of Biological Science, National Sun Yat-Sen University, Kaohsiung 80424, Taiwan; 9Department of Neurology, Xiamen Chang Gung Memorial Hospital, Xiamen 361126, China

**Keywords:** distal sensorimotor polyneuropathy, type 2 diabetes, Toronto Clinical Neuropathy Score, pulsatility index, carotid plaque score, composite amplitude scores, nerve conduction studies

## Abstract

Background: Diabetes contributes to a spectrum of complications encompassing microvascular and macrovascular disorders. This study aimed to explore the correlation between distal sensorimotor polyneuropathy (DSPN) severity and heightened carotid atherosclerosis among individuals with type 2 diabetes mellitus (T2DM)**.** Method: Participants underwent comprehensive assessments including nerve conduction studies (NCS), Toronto Clinical Neuropathy Score (TCNS) evaluations, assessment of cardiometabolic risk factors, and carotid sonography studies covering dynamic and morphological parameters. The resistance index (RI), pulsatility index (PI), peak systolic velocity (PSV), and end-diastolic velocity (EDV) in both the common carotid artery (CCA) and internal carotid artery (ICA), carotid intima-media thickness (IMT), and carotid plaque score (CPS) were also measured. Peripheral nerve function severity was assessed using composite amplitude scores (CAS) derived from NCS. Results: Individuals with DSPN exhibited lower EDV in the CCA and ICA (*p* < 0.0001 and *p* = 0.002), higher PI and RI in both CCA and ICA (all *p* < 0.0001), and higher CPS (*p* = 0.002). They also demonstrated a higher prevalence of retinopathy as an underlying condition, higher index HbA1c, and reduced estimated glomerular filtration rate (eGFR) (all *p* < 0.0001). Multiple linear regression analysis revealed significant associations where eGFR, ICA-PI, index HbA1c, waist circumference, and age were correlated with CAS. Meanwhile, diabetes duration, waist circumference, age, and index HbA1c showed significant associations with TCNS. Conclusions: Our study suggests that individuals with T2DM who exhibit more severe carotid atherosclerosis may not only be at increased risk of developing DSPN but also may experience greater severity of DSPN. PI in both the CCA and ICA, along with the CPS, serve as surrogate biomarkers for DSPN severity.

## 1. Introduction

Diabetes poses a significant global health challenge, precipitating a spectrum of complications encompassing microvascular disorders as well as macrovascular conditions [1,2,3]. Distal sensorimotor polyneuropathy (DSPN) is a common complication of microvascular disorders in type 2 diabetes (T2DM) [4]. It is characterized by progressive damage to the peripheral nerves which exhibits an upward trend commensurate with the duration of diabetes [5]. The advancement of DSPN may precipitate severe outcomes such as foot ulcers, infections, and, in severe cases, necessitate amputations, thereby contributing to escalated healthcare expenditures [6]. Atherosclerosis is a chronic inflammatory disease characterized by the accumulation of lipids, fibrous elements, and calcium in the arterial walls, leading to reduced elasticity and lumen narrowing [7]. In individuals with T2DM, the prevalence and severity of carotid atherosclerosis are more pronounced, increasing the risk of adverse outcomes and all-cause and cardiovascular mortality [8,9,10].

Carotid ultrasonography is a straightforward, non-invasive technique widely utilized for assessing atherosclerotic change [11]. The key indicators of atherosclerosis, namely carotid artery intima-media thickness (IMT), pulsatility index (PI), and resistance index (RI), can be obtained by ultrasound investigation of the carotid arteries. It has been demonstrated that these measurements serve as an independent predictor of cardiovascular events and mortality [12,13,14,15,16].

Although numerous studies have demonstrated a strong correlation between IMT and cardiovascular diseases, previous research has suggested that measurements of plaque area or volume may provide superior predictors of the inflammatory processes associated with atherosclerotic disease compared to intima-media thickness [17]. As a scoring system to assess carotid atherosclerosis, carotid plaque score (CPS) quantifies the extent of plaque accumulation, offering insight into the severity and potential clinical implications of carotid atherosclerosis [18]. CPS reflects the overall plaque burden in the carotid arteries and may be a more effective marker compared to IMT for evaluating macrovascular atherosclerotic change and predicting the recurrence of major adverse cardiovascular events (MACE) [19,20,21].

Emerging evidence suggests a notable association between the quantitative nerve conductive velocity and the carotid IMT in individuals with T2DM [22]. This relationship underscores the potential shared pathophysiological mechanisms, such as chronic inflammation, endothelial dysfunction, and oxidative stress, contributing to both neuropathy and carotid atherosclerosis. Although prior studies have examined the correlation between carotid atherosclerosis and polyneuropathy in T2DM, the interaction between DSPN severity and both the dynamic parameters and clinical morphological scores assessed by carotid ultrasonography, particularly through the evaluation of PI, RI, and CPS, has been insufficiently investigated. Moreover, previous research has frequently concentrated on IMT or pulse wave velocity without conducting a comprehensive evaluation of the entire spectrum of carotid atherosclerosis. The present study is novel in that it assesses these parameters simultaneously, thereby offering a more detailed characterization of macrovascular and microvascular complications in T2DM. The objective of this study is to address a significant knowledge gap regarding the reliability of predicting the severity of DSPN based on specific carotid atherosclerosis markers. The findings are expected to provide valuable clinical insights that could inform more personalized management strategies for patients with T2DM.

## 2. Patients and Methods

### 2.1. Study Subjects

This baseline cross-sectional analysis is based on our prospective cohort study. The prospective cohort study, conducted from April 2018 to June 2022 at a tertiary care center in southern Taiwan, included 298 patients diagnosed with T2DM. Patients were identified from hospital records and contacted during routine clinical visits. During screening, eligibility was confirmed through a thorough evaluation, including medical history, physical examination, and laboratory tests. The presence of DSPN was determined using the Toronto Consensus on Diabetic Neuropathy criteria, incorporating the Toronto Clinical Neuropathy Score (TCNS) and nerve conduction abnormalities, in line with the American Academy of Neurology guidelines [5,23,24].

Participants were excluded if they had neuropathies from non-diabetic causes, significant cardiovascular conditions unrelated to atherosclerosis, or a history of major cerebrovascular events or carotid surgery. After obtaining informed consent, participants completed a baseline assessment and returned to the clinic every three months for follow-up visits over a period of up to four years. The study included comprehensive data analysis at baseline and at 12-month intervals following clinical diagnosis, focusing on microvascular parameters, such as nerve conduction studies (NCS), and macrovascular parameters, such as carotid ultrasonography, to identify and monitor key outcomes. The study was approved by the hospital’s human research review committee (approval numbers 201800388B0, 201901363B0, and 202002095B0).

### 2.2. Clinical and Laboratory Measurements

All patients underwent thorough neurological and physical examinations administered by seasoned neurologists, encompassing initial clinical assessments and laboratory evaluations. The collected data encompassed demographic details including age at disease onset, sex, height, waist circumference, body mass index (BMI), disease duration, systolic and diastolic blood pressure, and microvascular complications related to diabetes. Each patient’s urinary albumin-to-creatinine ratio (UACR) [25] and estimated glomerular filtration rate (eGFR) [26] were also measured, as previously described. The TCNS questionnaire was administered to all participants, yielding a continuously variable range indicative of the severity of DSPN, ranging from 0 to 19 [27].

### 2.3. Assessment and Scoring of Nerve Conduction Studies

Nerve conduction studies (NCS) were performed by an experienced electrophysiologist using Nicolet Viking machines (Madison, WI, USA). Motor nerve studies included assessments of the median, ulnar, tibial, and peroneal nerves, while sensory nerve studies involved evaluations of the median, ulnar, and sural nerves. Parameters such as distal latency, amplitude, and nerve conduction velocity were measured for each nerve and compared against reference values established in our laboratory [28]. Sensory and motor nerves were bilaterally tested, with data recorded exclusively from the nerves on the non-dominant side for DSPN analysis.

Peripheral nerve function severity was evaluated using composite amplitude scores (CAS) derived from NCS. CAS encompassed measurements of peroneal compound muscle action potential (CMAP), tibial CMAP, ulnar CMAP, sural sensory nerve action potential (SNAP), and ulnar SNAP amplitudes. Points on a scale from 0 to 10 were assigned based on percentile values from our prior research, reflecting the severity across these five NCS attributes [29].

### 2.4. Evaluation of Carotid Atherosclerosis

Images were obtained using a B-mode ultrasound system (Philips HDI 5000 System, ATL-Philips, Bothell, WA, USA) equipped with a 4–10 MHz linear array transducer. The acquired images were then processed for automated measurements of carotid IMT using Q-LAB software from ATL-Philips on a workstation. Plaque was defined as localized wall thickening equal to or exceeding twice the thickness of the adjacent IMT [30]. Mean common carotid artery (CCA) IMT measurements were conducted by a single neurologist (D.Y.H.) using quantification software in a single-blind manner with respect to patients. The resistance index (RI), pulsatility index (PI), peak systolic velocity (PSV), and end-diastolic velocity (EDV) in both the common carotid artery (CCA) and internal carotid artery (ICA) were also measured.

The CPS was calculated by obtaining longitudinal images of the bilateral CCA, ICA, and carotid bulbs using a 90-degree rotated probe after initial transverse scans. Plaques were identified as focal intima-media thickening ≥1.1 mm without considering plaque length. The CPS summed the maximal thickness of plaques in specific segments: Segment 1 (ICA < 15 mm distal to CCA bifurcation), Segment 2 (ICA and CCA < 15 mm proximal to bifurcation), Segment 3 (CCA > 15 mm and <30 mm proximal to bifurcation), and Segment 4 (CCA > 30 mm proximal to bifurcation below flow divider) [31].

### 2.5. Statistical Analysis

Continuous variables were presented as mean ± SD or median (interquartile range). Logarithmic transformation was applied to variables that were not normally distributed to achieve normality. Differences in continuous variables between the two groups were assessed using independent *t*-tests. Correlation analysis investigated relationships between CAS and TCNS, along with variables such as age, disease duration, BMI, waist circumference, peripheral vascular risk factors, and carotid ultrasound parameters encompassing dynamic and morphological aspects, separately. Multiple linear regression analyses evaluated the impact of independent variables on CAS and TCNS individually, with variables selected based on significant correlations (*p* < 0.05) with CAS and TCNS, respectively. Data analysis was conducted using SPSS Statistics software (version 23, IBM; Redmond, WA, USA).

## 3. Results

### 3.1. Characteristics of Individuals with T2DM

The study enrolled 298 individuals diagnosed with T2DM, consisting of 140 women (mean age: 69.1 years) and 158 men (mean age: 69.6 years). Patient characteristics and underlying diseases at the last assessment are detailed in Table 1 and Table 2.

Compared to those without DSPN, patients with DSPN were older (*p* = 0.043), had longer diabetes duration (*p* < 0.001), had higher height, body weight, BMI, and waist circumference (*p* = 0.008, *p* = 0.001, *p* = 0.031, and *p* < 0.001, respectively), a higher prevalence of hyperlipidemia, coronary heart disease, and retinopathy as underlying diseases (*p* = 0.012, *p* = 0.001, and *p* < 0.001, respectively), higher HbA1c levels (*p* < 0.001), and lower eGFR (*p* = 0.028).

### 3.2. The Relationship between DSPN and Carotid Sonography Study

The association between DSPN and carotid sonography study was investigated in patients with T2DM, as presented in Table 3.

Individuals with DSPN exhibited lower EDV in the CCA and ICA (*p* < 0.0001 and *p* = 0.002), higher PI and RI in both CCA and ICA (all *p* < 0.0001), and higher carotid plaque scores (*p* = 0.002). However, there was no significant difference in PSV between patients with DSPN and those without DSPN in both CCA and ICA.

### 3.3. Impact of Carotid Atherosclerosis and Vascular Risk Factors on CAS and TCNS in Individuals with T2DM

The correlation analysis examining the impact of carotid atherosclerosis and other vascular risk factors on CAS and TCNS is summarized in Table 4. For CAS, significant correlations (correlation coefficient, *p*-value) were observed with age (r = 0.19, *p* < 0.001), diabetes duration (r = 0.34, *p* < 0.0001), BMI (r = 0.12, *p* = 0.045), waist circumference (r = 0.17, *p* = 0.004), HbA1c (r = 0.18, *p* = 0.002), UACR (r = 0.17, *p* = 0.005), eGFR (r = −0.36, *p* < 0.0001), CCA-EDV (r = −0.32, *p* < 0.0001), CCA-PI (r = 0.27, *p* < 0.0001), CCA-RI (r = 0.32, *p* < 0.0001), ICA-EDV (r = −0.21, *p* < 0.0001), ICA-PI (r = 0.29, *p* < 0.0001), ICA-RI (r = 0.27, *p* < 0.0001), and carotid plaque score (r = 0.22, *p* < 0.0001).

For TCNS, significant correlations were found with age (r = 0.19, *p* < 0.0001), disease duration (r = 0.37, *p* < 0.0001), BMI (r = 0.17, *p* = 0.0003), waist circumference (r = 0.25, *p* < 0.0001), HbA1c (r = 0.24, *p* < 0.001), UACR (r = 0.16, *p* = 0.009), eGFR (r = −0.24, *p* < 0.0001), CCA-EDV (r = −0.24, *p* < 0.0001), CCA-PI (r = 0.15, *p* = 0.008), CCA-RI (r = 0.16, *p* = 0.005), ICA-EDV (r = −0.15, *p* = 0.009), ICA-PI (r = 0.18, *p* = 0.002), ICA-RI (r = 0.18, *p* = 0.002), and carotid plaque score (r = 0.16, *p* = 0.007).

### 3.4. Association of Clinical Factors with CAS and TCNS in Individuals with T2DM

Only variables significantly correlated with CAS and TCNS, as detailed in Table 4, were included in the multiple linear regression analysis to identify the primary determinants influencing CAS and TCNS in individuals with T2DM.

In the multiple linear regression analysis, CAS and TCNS were used as dependent variables. Significant parameters from Table 4, including age, diabetes duration, BMI, waist circumference, index HbA1c, UACR, eGFR, mean CCA-EDV, mean CCA-PI, mean CCA-RI, mean ICA-EDV, mean ICA-PI, mean ICA-RI, and carotid plaque score, were incorporated as independent variables. The analysis revealed that eGFR, ICA-PI, HbA1c levels, waist circumference, and age were significantly associated with CAS, whereas diabetes duration, waist circumference, age, and HbA1c levels were significantly associated with TCNS (Table 5).

## 4. Discussion

### 4.1. Major Findings of Our Study

Our study substantiated the hypothesis of a notable correlation between the quantitative severity of peripheral nerve function, assessed through clinical and electrophysiological studies (e.g., CAS and TCNS), and the severity of carotid atherosclerosis, as evidenced by dynamic parameters (e.g., PI in both the CCA and ICA) and clinical morphological scores (e.g., CPS). Our study also underscores the robust association between microvascular and macrovascular complications in individuals with T2DM.

### 4.2. Association of Carotid Atherosclerosis Dynamic and Morphologic Parameters with DSPN Severity

Clinically, the PI serves as an indicator of distal vascular resistance and is employed to evaluate arterial narrowing as well as the effectiveness of treatments in conditions such as carotid artery stenosis [32]. Furthermore, the PI could be utilized as an additional measure for the evaluation of arterial wall shear stress, which is associated with an increased risk of cardiovascular disease and the development of atherosclerosis [33]. It is hypothesized that the elevated pulsations caused by the rigidity of macro-vasculature could influence capillary networks, thereby contributing to microvascular complications [34]. Previous research has demonstrated a positive correlation between increased PI and the development of microvascular complications, including renal dysfunction, retinopathy, and cerebral microangiopathy in patients with T2DM [35,36,37].

Our findings are consistent with prior research demonstrating an association between macrovascular disease and DSPN in diabetic populations. In T2DM, Kim et al. demonstrated a significant relationship between diabetic peripheral neuropathy and increased arterial stiffness without carotid intimal changes, and Avci et al. reported that diabetic peripheral neuropathy was significantly associated with both arterial stiffness and carotid IMT [38,39]. Tentolouris et al. suggested that increased aortic stiffness, measured by carotid-femoral pulse wave velocity, was significantly and independently linked to both the presence and severity of diabetic peripheral neuropathy in individuals with T2DM, regardless of existing risk factors [40]. Szczyrba et al. demonstrated that DSPN was associated with systemic macroangiopathy in individuals with type 1 diabetes mellitus (T1DM), as evidenced by increased carotid–femoral pulse wave velocity, but this association was not observed with cerebral macro- and microvascular impairment [41]. Previous research has indicated an elevated prevalence of polyneuropathy in individuals with diabetes and metabolic syndrome, both of which are prominent risk factors for the development of atherosclerosis [42,43]. Nevertheless, our study extends this knowledge by providing detailed quantitative assessments specifically linking carotid atherosclerosis with DSPN in T2DM patients. The present focus on pathological conditions affecting the carotid artery provides a novel perspective on the broader implications of vascular health on peripheral neuropathic conditions.

The existing literature has substantiated the intimate correlation between carotid plaque and the development of diabetic retinopathy and nephropathy [44,45]; however, the relationship between carotid plaque and DSPN remains inconsistent. Ferik et al. observed no statistically significant difference between DSPN and carotid atherosclerosis, defined by increased IMT and the presence of atherosclerotic plaque [46]. Cardoso et al. reported that the presence of plaques was predictive of the occurrence of cardiovascular events and renal outcomes, but not mortality or other microvascular complications, including peripheral neuropathy [47]. In contrast, a systematic review and meta-analysis revealed a significant correlation between carotid ultrasonographic parameters (IMT, carotid plaque, and carotid plaque score) and microvascular and macrovascular complications in individuals with diabetes [48]. Similarly, a study by Bartman et al. demonstrated that microvascular complications are independently associated with the carotid plaque score, but not with carotid IMT [49]. Consistent with prior research, the present study identified a significant association between CPS and the presence of DSPN in individuals with T2DM. However, no significant association was observed between carotid IMT and the presence of DSPN. The CPS serves as a scoring system for the assessment of carotid atherosclerosis by quantifying plaque accumulation and may be more precise than carotid IMT in reflecting the overall plaque burden. Consequently, CPS may offer a more effective marker than carotid IMT for the evaluation of macrovascular atherosclerotic changes [50], and is also linked to the severity of DSPN.

### 4.3. Pathophysiological Link between Carotid Atherosclerosis and DSPN Severity

The exact mechanisms underlying the association between carotid atherosclerosis and DSPN are complex and multifaceted. Diabetes complicates microvascular and macrovascular disease through impaired nitric oxide pathway function, disrupting vasorelaxation and promoting inflammation [51]. Systemic inflammation, a hallmark of atherosclerosis, likely plays a crucial role, with elevated inflammatory markers such as interleukin-6, tumor necrosis factor-α, and highly sensitive c-reactive protein contributing to oxidative stress and endothelial dysfunction [52]. This cascade can impair blood flow and nutrient delivery to peripheral nerves, resulting in ischemic injury and subsequent nerve degeneration. Furthermore, increased arterial stiffness can lead to microvascular dysfunction. Arterial stiffness impairs large artery buffering, increasing pressure and flow pulsatility. This pulsatility can damage microcirculation, inducing hypertrophic remodeling and vasoreactivity impairment. Arterial stiffening may contribute to fluctuations in blood pressure, which in turn may exacerbate microvascular dysfunction [53]. The generalized vascular pathology reflected by atherosclerotic change in the carotid arteries likely impacts the microvasculature supplying peripheral nerves. This study’s findings, including significant correlations between neuropathy severity (CAS and TCNS) and carotid ultrasonography parameters such as PI of CCA and ICA, and carotid plaque score, support the hypothesis that vascular health directly influences peripheral nerve function.

### 4.4. Associations of Risk Factors with DSPN Severity and Carotid Atherosclerosis

The significant association between carotid atherosclerosis and DSPN has important clinical implications. Our findings suggest that patients with carotid atherosclerosis should be routinely screened for DSPN, especially if they exhibit symptoms such as numbness, tingling, or extremity weakness. Early detection and management of polyneuropathy in these patients could improve their quality of life and prevent complications like falls, foot ulcers, and infections. Additionally, addressing common risk factors for atherosclerosis, such as hypertension, hyperlipidemia, and obesity, might also help mitigate the progression of DSPN [5]. Therapeutic strategies should include lifestyle modifications, and pharmacological interventions to control lipid levels and blood pressure. In the present study, HbA1c levels were identified as a significant factor associated with both CAS and TCNS. This finding suggests that, beyond HbA1c variability, sustained hyperglycemia may play a crucial role in the development of diabetic peripheral neuropathy [29,54,55].

### 4.5. Limitations of the Study

Although our study demonstrates a significant correlation between carotid atherosclerosis and DSPN severity in individuals with T2DM, it is important to note several limitations. Firstly, this study was a baseline cross-sectional analysis based on a prospective design, which remains limited in its capacity to definitively establish causality between carotid atherosclerosis and DSPN. To delineate the temporal sequence and causal connections more accurately between these conditions, further longitudinal analyses with extended follow-up periods are warranted. Secondly, the study population was restricted to older adults with pre-existing cardiovascular risk factors, potentially constraining the generalizability of our findings to younger age groups. The demographic profile of our sample, predominantly comprising older adults with a prolonged history of diabetes, may not fully reflect the broader population of individuals with T2DM. Future research should strive to encompass more diverse populations and investigate potential genetic and environmental factors that could impact the association between carotid atherosclerosis and DSPN. Finally, although treatment methods can influence the severity of DSPN and carotid atherosclerosis, our analysis revealed no statistically significant differences between the two patient groups regarding key variables, such as blood pressure and cholesterol levels. These factors were controlled in the statistical analysis to minimize their potential impact, ensuring that the observed associations are largely independent of ongoing treatments. However, it is acknowledged that different treatment regimens, including antihypertensive and lipid-lowering therapies, may affect vascular health and neuropathy progression. Future studies should incorporate detailed assessments of treatment types and intensities to evaluate their potential impact on the observed associations.

### 4.6. Clinical Significance of the Study

The findings emphasize the significant clinical observation that individuals with T2DM who demonstrate more pronounced carotid atherosclerosis may not only be at an elevated risk of developing DSPN but may also experience a more pronounced DSPN. This relationship suggests that the PI in both the CCA and ICA, as well as the CPS, can serve as surrogate biomarkers for assessing DSPN severity. The identification of these biomarkers is clinically significant because it allows for the early detection and more targeted management of patients at risk for severe DSPN, which may potentially improve patient outcomes by enabling timely interventions to prevent or mitigate neuropathic complications.

## 5. Conclusions

Our study suggests that individuals with T2DM who exhibit more severe carotid atherosclerosis may not only be at increased risk of developing DSPN but also may experience greater severity of DSPN. PI in both the CCA and ICA, along with the CPS, serve as surrogate biomarkers for DSPN severity.

## Figures and Tables

**Table 1 diagnostics-14-01922-t001:** Characteristics of patients with type 2 diabetes with and without Diabetic Sensorimotor Polyneuropathy.

	With DSPN (*n* = 81)	Without DSPN (*n* = 217)	*p*-Value
Characteristics			
Age (year)	70.9 ± 8.7	68.7 ± 7.9	0.043
Sex (male/female)	49/32	158/140	0.114
Diabetes duration (year)	14.0 ± 9.4	7.9 ± 8.0	<0.001
Height (cm)	163.3 ± 8.3	160.6 ± 7.5	0.008
Body weight (kg)	73.4 ± 14.6	68.0 ± 12.2	0.001
Body mass index (kg/m^2^)	27.5 ± 4.9	26.3 ± 4.1	0.031
Waist circumstance (cm)	96.7 ± 11.3	91.8 ± 10.5	<0.001
SBP (mmHg)	140.4 ± 20.0	137.9 ± 18.5	0.314
DBP (mmHg)	77.9 ± 11.3	77.8 ± 12.0	0.968
MAP (mmHg)	98.7 ± 13.6	97.8 ± 13.3	0.618
Pulse pressure (mmHg)	62.5 ± 12.2	60.1 ± 12.2	0.13
Baseline underlying disease			
Hypertension (%)	65 (80.2)	158 (74.2)	0.188
Hyperlipidemia (%)	33 (40.7)	124 (58.2)	0.012
Coronary heart disease (%)	12 (14.8)	9 (4.2)	0.001
Ischemic stroke (%)	22 (27.2)	46 (21.6)	0.245
Peripheral artery disease (%)	5 (6.2)	7 (3.3)	0.257
Retinopathy (%)	23 (28.4)	23 (10.8)	<0.001

Data are presented as means ± standard deviations or *n* (%). Abbreviations: *n*, number of cases; DSPN, distal sensorimotor polyneuropathy; SBP, systolic blood pressure; DBP, diastolic blood pressure; MAP, mean arterial pressure.

**Table 2 diagnostics-14-01922-t002:** Laboratory test findings of patients with type 2 diabetes with and without Diabetic Sensorimotor Polyneuropathy.

	With DSPN (*n* = 81)	Without DSPN (*n* = 217)	*p*-Value
Laboratory test findings			
Total cholesterol(mmol/L)	159.1 ± 30.2	159.8 ± 28.2	0.867
Triglyceride(mmol/L)	136.9 ± 76.6	139.3 ± 77.9	0.809
HDL-C (mmol/L)	47.6 ± 14.1	49.2 ± 15.6	0.437
LDL-C (mmol/L)	91.7 ± 37.0	87.3 ± 33.7	0.329
Index HbA1c (%)	7.6 ± 1.4	7.0 ± 0.7	<0.0001
UACR (mg/g)	29 (10.4, 124.1)	12.9 (5.9, 38.2)	0.052
eGFR (mL/min/1.73 m^2^)	56.2 ± 26.9	76.8 ± 25.9	<0.0001

Data are presented as means ± standard deviations or *n* (%) or median with interquartile range. Abbreviations: *n*, number of cases; DSPN, distal sensorimotor polyneuropathy; HDL-C, high-density lipoprotein cholesterol; LDL-C, low-density lipoprotein cholesterol; HbA1c, glycohemoglobin; eGFR, estimated glomerular filtration rate; UACR, urine albumin-creatinine ratio.

**Table 3 diagnostics-14-01922-t003:** NCS, TCNS, and carotid sonography study in patients with type 2 diabetes.

	With DSPN (*n* = 81)	Without DSPN (*n* = 217)	*p*-Value
NCS			
Composite amplitude scores	6.1 ± 2.3	1.9 ± 1.8	<0.0001
Median nerve, motor			
CMAP, left (mV)	8.2 ± 2.7	9.4 ± 2.4	<0.0001
MNCV	49 ± 4.9	54 ± 4.0	<0.0001
Ulnar nerve, motor			
CMAP, left (mV)	8.2 ± 2.1	9.3 ± 2.0	<0.0001
MNCV	50.0 ± 5.2	54.4 ± 5.0	<0.0001
Peroneal nerve, motor			
CMAP, left (mV)	3.3 ± 2.6	4.7 ± 2.2	<0.0001
MNCV	40.0 ± 3.9	45.9 ± 3.8	<0.0001
Tibial nerve			
CMAP, left (mV)	6.2 ± 4.4	11.7 ± 4.3	<0.0001
MNCV	39.7 ± 4.3	46.0 ± 4.0	<0.0001
Ulnar nerve, sensory			
SNAP, left (µV)	15.9 ± 12.5	29.2 ± 14.1	<0.0001
SNCV	46.7 ± 11.0	54.4 ± 6.2	<0.0001
Sural nerve, sensory			
SNAP, left (µV)	2.6 ± 2.3	11.4 ± 5.1	<0.0001
SNCV	42.2 ± 5.6	50.6 ± 5.8	<0.0001
TCNS	10.4 ± 3.0	4.6 ± 2.2	<0.0001
Carotid sonography			
Mean IMT	0.09 ± 0.03	0.09 ± 0.06	0.534
Mean CCA-PSV	82.6 ± 20.5	83.7 ± 20.2	0.664
Mean CCA-EDV	16.3 ± 5.0	20.0 ± 5.6	<0.0001
Mean CCA-PI	1.9 ± 0.5	1.6 ± 0.4	<0.0001
Mean CCA-RI	0.8 ± 0.1	0.8 ± 0.1	<0.0001
Mean ICA-PSV	67.2 ± 19.5	66.8 ± 15.2	0.855
Mean ICA-EDV	20.7 ± 5.8	23.3 ± 6.2	0.002
Mean ICA-PI	1.2 ± 0.3	1.1 ± 0.2	<0.0001
Mean ICA-RI	0.7 ± 0.1	0.6 ± 0.1	<0.0001
Carotid plaque score	8.1 ± 6.3	5.6 ± 5.5	0.002

Data are presented as means ± standard deviations or median (IQR) *n* (%). Abbreviations: IQR = interquartile range; *n*, number of cases; DSPN, distal sensorimotor polyneuropathy; NCS, nerve conduction study; CMAP, compound muscle action potential; SNAP, sensory nerve action potential; MNCV, motor nerve conduction velocity; SNCV, sensory nerve conduction velocity; TCNS, Toronto Clinical Neuropathy Score; IMT, intima-media thickness; RI, resistance index; PI, pulsatility index; PSV, peak systolic velocity; EDV, end-diastolic velocity, CCA, common carotid artery; ICA, internal carotid artery.

**Table 4 diagnostics-14-01922-t004:** Correlation analysis of CAS and TCNS in patients with type 2 diabetes.

Variables	CAS		TCNS	
	r	*p* Value	r	*p* Value
Age (year)	0.191	<0.0001	0.191	<0.0001
Diabetes duration (year)	0.338	<0.0001	0.367	<0.0001
Body mass index (kg/m^2^)	0.116	0.045	0.169	0.003
Waist circumstance (cm)	0.168	0.004	0.253	<0.0001
MAP (mmHg)	0.056	0.333	0.062	0.288
Pulse pressure (mmHg)	0.082	0.159	0.061	0.296
Total cholesterol (mmol/L)	−0.035	0.553	−0.017	0.77
Triglyceride (mmol/L)	−0.018	0.763	0.028	0.636
HDL-C (mmol/L)	−0.056	0.343	−0.019	0.745
LDL-C (mmol/L)	0.009	0.884	−0.076	0.199
Index HbA1c	0.182	0.002	0.242	<0.0001
UACR (mg/g)	0.173	0.005	0.162	0.009
eGFR (mL/min/1.73 m^2^)	−0.358	<0.0001	−0.235	<0.0001
Mean IMT	−0.004	0.94	0.006	0.924
Mean CCA-PSV	−0.036	0.541	−0.089	0.13
Mean CCA-EDV	−0.315	<0.0001	−0.242	<0.0001
Mean CCA-PI	0.269	<0.0001	0.154	0.008
Mean CCA-RI	0.324	<0.0001	0.163	0.005
Mean ICA-PSV	0.022	0.706	0.002	0.967
Mean ICA-EDV	−0.208	<0.0001	−0.152	0.009
Mean ICA-PI	0.286	<0.0001	0.182	0.002
Mean ICA-RI	0.265	<0.0001	0.181	0.002
Carotid plaque score	0.215	<0.0001	0.157	0.007

r: correlation coefficient. Abbreviations: CAS, composite amplitude scores; TCNS, Toronto Clinical Neuropathy Score; MAP, mean arterial pressure; HDL-C, high-density lipoprotein cholesterol; LDL-C, low-density lipoprotein cholesterol; HbA1c, glycohemoglobin; eGFR, estimated glomerular filtration rate; UACR, urine albumin-creatinine ratio; IMT, intima-media thickness; RI, resistance index; PI, pulsatility index; PSV, peak systolic velocity; EDV, end-diastolic velocity, CCA, common carotid artery; ICA, internal carotid artery.

**Table 5 diagnostics-14-01922-t005:** Effects of the variables on CAS and TCNS in patients with type 2 diabetes according to correlation analysis.

	Regression Coefficient	Standard Error	*p* Value	95% Confidence Interval
CAS				
Constant	−6.47	2.2	0.004	−10.82, −2.13
eGFR (mL/min/1.73 m^2^)	−0.025	0.006	<0.0001	−0.038, −0.013
Mean ICA-PI	1.86	0.62	0.003	0.64, 3.08
Index HbA1c	0.409	0.15	0.007	0.11, 0.71
Waist circumstance (cm)	0.033	0.013	0.014	0.007, 0.06
Age (year)	0.045	0.02	0.044	0.005, 0.084
TCNS				
Constant	−13.304	2.96	<0.0001	−19.14, −7.47
Diabetes duration (year)	0.115	0.026	<0.0001	0.064, 0.166
Waist circumstance (cm)	0.081	0.02	<0.0001	0.042, 0.12
Age (year)	0.098	0.027	<0.0001	0.046, 0.15
Index HbA1c	0.626	0.226	0.006	0.182, 1.07

The regression coefficient is presented for each individual variable. Abbreviations: CAS, composite amplitude score; TCNS, Toronto Clinical Neuropathy Score; PI, pulsatility index; ICA, internal carotid artery; HbA1c, glycohemoglobin; eGFR, estimated glomerular filtration rate.

## Data Availability

The data from this study can be acquired from the corresponding author upon reasonable request.

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
