# Peer review of "The Association between the Severity of Distal Sensorimotor Polyneuropathy and Increased Carotid Atherosclerosis in Individuals with Type 2 Diabetes"

_diagnostics, 2024, doi:10.3390/diagnostics14171922_

Round 1
Reviewer 1 Report
Comments and Suggestions for Authors
This is an exciting research paper.
However, a few suggestions are placed to further improve the manuscript.
Introduction:
Comment : The introduction fails to set in the need of the study. The aims and objective for the study should be matching with the final conclusion irrespective of the positive or the negative findings of the study. Hence, kindly mention, wether similar study been done before ? If yes, what extra, we are achieving by this study ?
Method:
Comment : Hen was the study conducted. It is important to know the time line of start and completion of the study. In this prospective study, the one line exclusion criteria does not make any sense. The exclusion criteria needs to be more elaborative reflective of the aims of the study.
Results:
Comment : The data has been nicely analysed and hence the results are also very elaborative. However, the short form/ acronyms in the tables are sometimes difficult to understand. The full forms of every short form/ acronyms must be mentioned with the tables.
Discussion:
Comment : First paragraph of discussion depicting major findings has been nicely written. However, in the second paragraph, the emphasis must be on the neuropathy and atherosclerosis (as per the aim of the study) and not on atherosclerosis and cardiovascular risks. The same thing may be written later on (if authors want).
Comments on the Quality of English Languageminor grammatical errors
Author Response
Reviewer 1)
Comments and Suggestions for Authors
This is an exciting research paper. However, a few suggestions are placed to further improve the manuscript.
- Introduction: Comment: The introduction fails to set in the need of the study. The aims and objective for the study should be matching with the final conclusion irrespective of the positive or the negative findings of the study. Hence, kindly mention, whether similar study been done before? If yes, what extra, we are achieving by this study ?
Answer: Thanks for your suggestion, we have adjusted the last paragraph of the introduction section as follows:
Although prior studies have examined the correlation between carotid atherosclerosis and polyneuropathy in T2DM, the interaction between DSPN severity and both the dynamic parameters and clinical morphological scores assessed by carotid ultrasonography, particularly through the evaluation of PI, RI, and CPS, has been insufficiently investigated. Moreover, previous research has frequently concentrated on IMT or pulse wave velocity without conducting a comprehensive evaluation of the entire spectrum of carotid atherosclerosis. The present study is novel in that it assesses these parameters simultaneously, thereby offering a more detailed characterization of macrovascular and microvascular complications in T2DM. The objective of this study is to address a significant knowledge gap regarding the reliability of predicting the severity of DSPN based on specific carotid atherosclerosis markers. The findings are expected to provide valuable clinical insights that could inform more personalized management strategies for patients with T2DM.
- Method: Comment: Hen was the study conducted. It is important to know the time line of start and completion of the study. In this prospective study, the one-line exclusion criteria do not make any sense. The exclusion criteria need to be more elaborative reflective of the aims of the study.
Answer: Thanks for your suggestion, we have adjusted the first paragraph of the Method section as follows:
This prospective cohort study, conducted from April 2018 to June 2022 at a tertiary care center in southern Taiwan, included 298 patients diagnosed with T2DM. Patients were identified from hospital records and contacted during routine clinical visits. During screening, eligibility was confirmed through a thorough evaluation, including medical history, physical examination, and laboratory tests. The presence of DSPN was determined using the Toronto Consensus on Diabetic Neuropathy criteria, incorporating the Toronto Clinical Neuropathy Score (TCNS) and nerve conduction abnormalities, in line with the American Academy of Neurology guidelines [5,23,24]. Participants were excluded if they had neuropathies from non-diabetic causes, significant cardiovascular conditions unrelated to atherosclerosis, or a history of major cerebrovascular events or carotid surgery. After obtaining informed consent, participants underwent baseline assessments, including nerve conduction studies (NCS) and carotid ultrasonography, with outcomes related to DSPN and carotid atherosclerosis monitored prospectively. The study was approved by the hospital's human research review committee (approval numbers 201800388B0, 201901363B0, and 202002095B0).
- Results: Comment: The data has been nicely analyzed and hence the results are also very elaborative. However, the short form/ acronyms in the tables are sometimes difficult to understand. The full forms of every short form/ acronym must be mentioned with the tables.
Answer: Thank you for your thoughtful reminder. We have modified some abbreviations.
- Discussion: Comment: First paragraph of discussion depicting major findings has been nicely written. However, in the second paragraph, the emphasis must be on the neuropathy and atherosclerosis (as per the aim of the study) and not on atherosclerosis and cardiovascular risks. The same thing may be written later on (if authors want).
Answer: Thank you for your valuable comment. In the second paragraph, we emphasize the relationship between neuropathy and atherosclerosis, particularly focusing on both dynamic and morphological parameters. Specifically, we discuss morphological markers such as carotid IMT and CPS, highlighting that in our study, CPS demonstrates a significant correlation with DSPN severity (as measured by CAS and TCNS) in comparison to carotid IMT. Based on these findings, we conclude that CPS may be a more effective marker than carotid IMT for evaluating macrovascular atherosclerotic changes and is also associated with the severity of DSPN. To clarify this point, we have revised the sentence as follows:
Consequently, CPS may offer a more effective marker than carotid IMT for the evaluation of macrovascular atherosclerotic changes [50], and is also linked to the severity of DSPN.
Reviewer 2 Report
Comments and Suggestions for Authors
Authors aimed to investigate the potential association between the severity of DSPN and increased carotid atherosclerosis in individuals with T2DM. The study appears interesting, but the following issues needs to be clarified.
Why was no controls recruited for the study ? That will give the Baseline data for NCS and CS.
Why do authors mention the study as observational, prospective cohort analysis?
Details regarding Patient recruitment have to be mentioned in detail.
Treatment status of patient ? How does the treatment modality affect the experiment?
Study period to be mentioned
Table 5 is not cited in results
Table 5 do not list all the variables that were significant in the Coorelation analysis. Please include all paramters . If authors have used different models for regression. Please specify that also.
Regression coeff is highest for Index HbA1c for TCNS. Please comment on the significance.
Please mention what could be the potential confounders in the study.
What is the clinical significance of the study ?
Author Response
Reviewer 2)
Comments and Suggestions for Authors
Authors aimed to investigate the potential association between the severity of DSPN and increased carotid atherosclerosis in individuals with T2DM. The study appears interesting, but the following issues needs to be clarified.
- Why were no controls recruited for the study? That will give the Baseline data for NCS and CS.
Answer: Thanks for your suggestion. The study did not include a control group because the primary aim was to investigate the relationship between the severity of distal sensorimotor polyneuropathy (DSPN) and carotid atherosclerosis in individuals with type 2 diabetes mellitus (T2DM). Instead of recruiting a separate control group, we utilized established reference values for nerve conduction studies (NCS) from our laboratory, which have been validated in previous studies. These reference values provide a reliable baseline for assessing the parameters such as distal latency, amplitude, and nerve conduction velocity, allowing us to make meaningful comparisons within the diabetic population. - Why do authors mention the study as observational, prospective cohort analysis?
Answer: Thank you for your suggestion. The term "observational, prospective cohort analysis" is used to describe a study that involves observing a group of individuals (cohort) over a period of time without any intervention from the researchers, which is a typical feature of observational studies. In our study, we prospectively followed participants, beginning data collection at specific time points and observing outcomes as they occurred. This approach allowed us to examine the association between exposures (e.g., severity of DSPN) and outcomes (e.g., carotid atherosclerosis) within defined cohorts.
- Details regarding Patient recruitment have to be mentioned in detail.
Answer: Thank you for your suggestion. we have adjusted the first paragraph of the Method as follows
This prospective cohort study, conducted from April 2018 to June 2022 at a tertiary care center in southern Taiwan, included 298 patients diagnosed with T2DM. Patients were identified from hospital records and contacted during routine clinical visits. During screening, eligibility was confirmed through a thorough evaluation, including medical history, physical examination, and laboratory tests. The presence of DSPN was determined using the Toronto Consensus on Diabetic Neuropathy criteria, incorporating the Toronto Clinical Neuropathy Score (TCNS) and nerve conduction abnormalities, in line with the American Academy of Neurology guidelines [5,23,24].
Participants were excluded if they had neuropathies from non-diabetic causes, significant cardiovascular conditions unrelated to atherosclerosis, or a history of major cerebrovascular events or carotid surgery. After obtaining informed consent, participants underwent baseline assessments, including nerve conduction studies (NCS) and carotid ultrasonography, with outcomes related to DSPN and carotid atherosclerosis monitored prospectively. The study was approved by the hospital's human research review committee (approval numbers 201800388B0, 201901363B0, and 202002095B0).
- Treatment status of patient? How does the treatment modality affect the experiment?
Answer: Thank you for your suggestion. Because the blood pressure and blood lipid control of the two groups of patients were similar, we did not discuss this. The relevant content will be mentioned in the Limitation.
Although treatment methods can influence the severity of DSPN and carotid atherosclerosis, our analysis revealed no statistically significant differences between the two patient groups regarding key variables, such as blood pressure and cholesterol levels. These factors were controlled in the statistical analysis to minimize their potential impact, ensuring that the observed associations are largely independent of ongoing treatments. However, it is acknowledged that different treatment regimens, including antihypertensive and lipid-lowering therapies, may affect vascular health and neuropathy progression. Future studies should incorporate detailed assessments of treatment types and intensities to evaluate their potential impact on the observed associations.
- Study period to be mentioned
Answer: Thank you for your suggestion. we have adjusted the first paragraph of the Method section as follows
This prospective cohort study, conducted from April 2018 to June 2022 at a tertiary care center in southern Taiwan, included 298 patients diagnosed with T2DM.
- Table 5 is not cited in results
Answer: Thank you for your thoughtful reminder. We cited Table 5 in the part 3.4 of the Result section.
The analysis revealed that eGFR, ICA-PI, HbA1c levels, waist circumference, and age were significantly associated with CAS, whereas diabetes duration, waist circumference, age, and HbA1c levels were significantly associated with TCNS (Table 5).
- Table 5 do not list all the variables that were significant in the correlation analysis. Please include all paramters. If authors have used different models for regression. Please specify that also.
Answer: Thank you for your insightful comment. In response, we included all variables that were significantly correlated with CAS and TCNS in the multiple linear regression model, analyzing them separately. To clarify this approach, we have added the following details:
In the multiple linear regression analysis, CAS and TCNS were used as dependent variables. Significant parameters from Table 4, including age, diabetes duration, BMI, waist circumference, index HbA1c, UACR, eGFR, mean CCA-EDV, mean CCA-PI, mean CCA-RI, mean ICA-EDV, mean ICA-PI, mean ICA-RI, and carotid plaque score, were incorporated as independent variables.
- Regression coefficient is highest for Index HbA1c for TCNS. Please comment on the significance.
Answer: Thank you for your suggestion. We added some explanation in part 4.4 of Discussion section as follows:
In the present study, HbA1c levels were identified as a significant factor associated with both CAS and TCNS. This finding suggests that, beyond HbA1c variability, sustained hyperglycemia may play a crucial role in the development of diabetic peripheral neuropathy.
- Please mention what could be the potential confounders in the study.
Answer: Thank you for your suggestion. We have adjusted part of the Limitation paragraph as follows:
Firstly, despite employing a prospective design, this study remains limited in its capacity to definitively establish causality between carotid atherosclerosis and DSPN. To delineate the temporal sequence and causal connections more accurately between these conditions, further longitudinal analyses with extended follow-up periods are warranted.
- What is the clinical significance of the study?
Answer: Thank you for your suggestion. We added part 4.6 to the Discussion section.
The findings emphasize the significant clinical observation that individuals with T2DM who demonstrate more pronounced carotid atherosclerosis may not only be at an elevated risk of developing DSPN but may also experience a more pronounced DSPN. This relationship suggests that the PI in both the CCA and ICA, as well as the CPS, can serve as surrogate biomarkers for assessing DSPN severity. The identification of these biomarkers is clinically significant because it allows for the early detection and more targeted management of patients at risk for severe DSPN, which may potentially improve patient outcomes by enabling timely interventions to prevent or mitigate neuropathic complications.
Round 2
Reviewer 2 Report
Comments and Suggestions for Authors
1. Please include the established reference values for nerve conduction studies (NCS) from the laboratory in the tables.
2. Authors mention that they have prospectively followed participants, beginning data collection at specific time points and observing outcomes as they occurred. Please describe ( in detail ) in methodology, the time duration of follow-up and the outcomes that have been observed at the end of follow-up.
Author Response
Mr. Dusan Vukelic,
Assistant Editor, MDPI Novi Sad
Email: dusan.vukelic@mdpi.com
MDPI Branch Office, Novi Sad
Bulevar oslobođenja 83, 21000 Novi Sad, Serbia
Tel.: +381 21 300 14 49
Dear Mr. Dusan Vukelic:
Greetings!
Thank you for your valuable suggestions regarding our manuscript (Manuscript ID: diagnostics-3135092 Type of manuscript: Article Title: The Association Between the Severity of Distal Sensorimotor Polyneuropathy and Increased Carotid Atherosclerosis in Individuals with Type 2 Diabetes). We have made appropriate changes and included substantial improvements in the revised manuscript. Following are our point-by-point responses to the reviewers’ comments.
We hope that these changes concur with your comments and adequately address the concerns raised. We also hope that this paper will be re-evaluated for publication in its improved version.
Thank you very much. We look forward to hearing from you soon.
Sincerely,
Cheng-Hsien Lu, M.D., M.Sc.
Department of Neurology
Kaohsiung Chang Gung Memorial Hospital
123, Ta Pei Road, Niao Sung,
Kaohsiung 83304, Taiwan
Tel: +886-7-7317123 ext.2283
E-mail: chlu99@ms44.url.com.tw and chlu99@adm.cgmh.org.tw
Comments and Suggestions for Authors
- Please include the established reference values for nerve conduction studies (NCS) from the laboratory in the tables.
Answer: Thank you for your valuable suggestion. We understand your request to include the reference values for nerve conduction studies (NCS) in the tables. However, in Table 3, our primary focus is on comparing the differences between the groups with and without diabetic neuropathy. Including the reference values in the main table may distract from this focus. After extensive discussions with all co-authors, we have decided to retain the current format of Table 3. Additionally, we have provided the relevant reference for your review and have included the standard reference values used in our laboratory within the Methods section for the reader's benefit.
The reference values ​​we use have been previously published and we provide you with these reference values ​​and related publications here. Thank you for your understanding and support of our work.
|
|
P-value |
|
NCS |
Normal reference value |
|
Median nerve, motor |
|
|
CMAP, left (mV) |
5.2-17.2 |
|
MNCV |
52-65 |
|
Ulnar nerve, motor |
|
|
CMAP, left (mV) |
6.0-17.3 |
|
MNCV |
52-69 |
|
Peroneal nerve, motor |
|
|
CMAP, left (mV) |
2.5-12.5 |
|
MNCV |
42-55 |
|
Tibial nerve |
|
|
CMAP, left (mV) |
6.4-28.5 |
|
MNCV |
41-57 |
|
Ulnar nerve, sensory |
|
|
SNAP, left (µV) |
11.1-83.9 |
|
SNCV |
46-67 |
|
Sural nerve, sensory |
|
|
SNAP, left (µV) |
5.8-42.2 |
|
SNCV |
38-56 |
Huang CR, Chang WN, Chang HW, Tsai NW, Lu CH. Effects of age, gender, height, and weight on late responses and nerve conduction study parameters. Acta Neurol Taiwan. 2009;18(4):242-9. PubMed PMID: 20329591.
- Authors mention that they have prospectively followed participants, beginning data collection at specific time points and observing outcomes as they occurred. Please describe (in detail) in methodology, the time duration of follow-up and the outcomes that have been observed at the end of follow-up.
Answer: Thank you for your feedback. We acknowledge the need for a more detailed description of the follow-up duration and the outcomes observed at the end of the study. In the revised methodology section, we will include the following details and add the revised sentences in method section. They are as follows:
This prospective observational study, conducted from April 2018 to June 2022 at a tertiary care center in southern Taiwan, included 298 patients diagnosed with T2DM. DSPN was confirmed according to Toronto Consensus on Diabetic Neuropathy criteria. Participants were excluded if they had neuropathies from non-diabetic causes, significant cardiovascular conditions unrelated to atherosclerosis, or a history of major cerebrovascular events or carotid surgery.
After obtaining informed consent, participants completed a baseline assessment and returned to the clinic every three months for follow-up visits over a period of up to four years. The study included comprehensive data analysis at baseline and at 12-month intervals following clinical diagnosis, focusing on microvascular parameters, such as nerve conduction studies (NCS), and macrovascular parameters, such as carotid ultrasonography, to identify and monitor key outcomes. The study was approved by the hospital's Human Research Review Board (approval numbers 201800388B0, 201901363B0, and 202002095B0)
